# OpenReview forum: "Adaptive Tensor Attention Networks with Cross-Domain Transfer for Drug-Target Interaction Prediction"
_ICLR.cc/2025/Conference — Submitted to ICLR 2025_

### Official Review · Reviewer_UrG2 · 2024-10-29

**Soundness:** 2
**Presentation:** 2
**Contribution:** 2
**Rating:** 3
**Confidence:** 5

**Summary:**

The authors present a new framework for Drug-Target Interaction with Domain Adaptation (DTI-DA) by adopting concepts from multiple fields, including quantum mechanics, differential geometry, and information theory. This approach introduces a DTI symplectic structure to capture the drug-target interactions, which finally leads to a Quantum Optimal Transport problem that supports domain adaptation. Empirical results demonstrate improvements in DTI prediction accuracy on BindingDB and BioSNAP datasets compared to the existing methods.

**Strengths:**

1.	The framework incorporates multiple disciplines, including quantum mechanics, information theory, and statistical learning to formulate the DTI domain adaption problem.
2.	The paper derives several theorems, such as the Quantum Optimal Transport theorem, which supports the cross-domain adaptation in DTI.
3.	The model achieves notable performance improvements in metrics like AUC and AUPR in several different datasets compared to a few baseline models.

**Weaknesses:**

1.	The integration of too much theory makes it challenging for readers without a strong background in these areas to follow the derivations and get the key points from the theoretical part.
2.	Benchmark results are not as comprehensive as the theory part. Different settings should be explored, such as blind test on drug-protein pair where drug and protein are not seen in the training data.
3.	DTI prediction has been extensively studies. Many baselines are missed in the paper, such as DeepPurpose.
4.	The application to real-world drug discovery pipelines is not explored. More validation exploration could strengthen the practical impact of the approach

**Questions:**

1.	Simplifying some of the theoretical explanations or providing intuitive interpretations could improve the readability of the paper.
2.	Many baselines are missed. Please do a comprehensive literature review of the existing DTI prediction methods and include the state-of-the-arts methods for comparison.
3.	How interpretable are the model’s predictions or the model’s parameters, particularly in the quantum setting? Can the model provide any insights into the underlying chemical or biological mechanisms of drug-target interactions?
4.	Scalability: How does the proposed method handle large-scale datasets, particularly in terms of computational efficiency?

---

### Official Review · Reviewer_NQfK · 2024-11-01

**Soundness:** 3
**Presentation:** 2
**Contribution:** 2
**Rating:** 5
**Confidence:** 3

**Summary:**

The authors introduce a comprehensive unified theory for Drug-Target Interaction prediction with Domain Adaptation (DTI-DA), integrating quantum mechanics, differential geometry, and information theory. The authors develop a novel framework use a DTI symplectic structure that captures molecular interaction geometry and a quantum statistical mechanical formulation incorporating DTI-preserving quantum features.
The implementation architecture consists of two main components: a Graph Attention Network (GAT) for adaptive neighbor attention and a Knowledge-Aware Network (KAN) for structural optimization. Experimental validation was conducted on two datasets: BioSNAP and BindingDB.

On the BioSNAP dataset DTI-DA model achieved an AUC of 0.744 (2.66% improvement over baseline), AUPR of 0.757 (3.42% improvement), and accuracy of 0.659 (3.71% improvement). Performance on the BindingDB dataset was also strong, with improvements of 8.11% and 9.075% over SVM and RF baselines respectively.

**Strengths:**

* Comprehensive theoretical foundation combining multiple mathematical frameworks (quantum mechanics, differential geometry, information theory) in a novel way
* Mathematical proofs and derivations, particularly in sections 2.1-2.4

**Weaknesses:**

* Limited ablation studies and theoretical complexity may limit practical implementation.
* Experiments are conducted with just datasets. An interesting benchmark would be Merck FEP (https://chemrxiv.org/engage/chemrxiv/article-details/60c747cc469df43efff438b9)

**Questions:**

* What is the computational complexity of implementing the quantum optimal transport?
* How does the model handle low precision numerics in quantum channel calculations?
   * For example, if someone uses mixed precision training (bf16/fp32) instead of fp32.
* Please fix the formatting issue with the references. Some references have a white spaces and some don't have it.
  * Knowledge Aware Network's reference on L134 is wrong.

---

### Official Review · Reviewer_kYsB · 2024-11-03

**Soundness:** 1
**Presentation:** 1
**Contribution:** 1
**Rating:** 1
**Confidence:** 4

**Summary:**

The paper is nonsensical and in all likelihood LLM written.

It cites random papers. Its statements are disconnected. The equations are nonsensical. The experiments are nonsensical.

I tried 3 LLM detection tools and all classified the paper as LLM generated.

**Strengths:**

NA

**Weaknesses:**

NA

**Questions:**

NA

---

### Official Review · Reviewer_wwZV · 2024-11-04

**Soundness:** 2
**Presentation:** 1
**Contribution:** 1
**Rating:** 1
**Confidence:** 4

**Summary:**

This work purported to introduce a "groundbreaking unified theory for Drug-Target Interaction prediction with Domain Adaptation (DTI-DA) that bridges the gap between quantum mechanics, information theory, and statistical learning". To achieve this, it proposed to use a novel symplectic structure that captures the information geometry of DTIs. While this appears to be an interesting direction, the main text consists mostly of derivations of quantities without context. It is also unclear how the derivations relate the application that the manuscript sought to address, which is DTI. Perhaps the manuscript did make some important breakthroughs but these have unfortunately not shown with the text provided.

**Strengths:**

- The overall idea relating symplectic structures with quantum optimal transport appears to be novel.

**Weaknesses:**

- The presentation does not make it clear how the "unified variational principle" was used in model training or how it was related to the model architecture (Figure 1), which also uses very "antiquated" modules (GAT and KAN).
- The experiments were far from adequate: there should have been multiple runs (for uncertainty quantification) and more "modern" baselines to compare with.
- Typo in Figure 2: the $ x $-labels should have been in the other direction.

**Questions:**

N/A

---

### Meta-Review · Area_Chair_ZLhs · 2024-12-17

**Metareview:**

All reviewers mentioned several weaknesses and problems, and none of them assigned a positive score. The most critical remarks concerned the unclear connection of the theory parts with real-world applications, problems regarding model (or parameter) interpretation, unclear scalability and missing benchmark experiments. There was no rebuttal, so I recommend rejection of this paper.

**Additional Comments On Reviewer Discussion:**

There are many open questions, but no rebuttal...

---

### Decision · Program_Chairs · 2025-01-22

Reject